# Effectiveness of Drama-Based Intervention in Improving Mental Health and Well-Being: A Systematic Review and Meta-Analysis during the COVID-19 Pandemic and Post-Pandemic Period

**DOI:** 10.3390/healthcare11060839

**Published:** 2023-03-13

**Authors:** Lulu Jiang, Farideh Alizadeh, Wenjing Cui

**Affiliations:** 1Faculty of Creative Arts, University of Malaya, Kuala Lumpur 50603, Malaysia; lulu.jiang@hotmail.com; 2Drama Department, Faculty of Creative Arts, University of Malaya, Kuala Lumpur 50603, Malaysia; 3Department of Health Management Centre, Qilu Hospital, Cheeloo College of Medicine, Shandong University, Jinan 250012, China; cuiwenjing@qiluhospital.com

**Keywords:** systematic review, meta-analysis, drama-based intervention, mental health, well-being, COVID-19

## Abstract

As a creative form of psychotherapy, drama appears to assist individuals in the COVID-19 pandemic and post-pandemic period in altering crisis conditions and challenging negative perspectives. Drama-based intervention is presented as an option for addressing mental health issues in clinical and general populations by utilising various multidisciplinary sources, such as psychodrama and role playing. In this study, a systematic review and meta-analysis were employed to assess the impact of drama on mental health and well-being during the COVID-19 pandemic and post-pandemic. Four electronic databases (PubMed, Cochrane, Web of Science, and ScienceDirect) were extensively searched from December 2019 to October 2022. Quality assessment and Risk of Bias tool of the Cochrane Collaboration were performed. Using a random effect model, standardised mean difference (SMD) values and 95% confidence intervals (CI) were calculated. In the final analysis, 25 studies involving 797 participants were included. The study revealed that drama-based interventions have the potential to improve mental health (e.g., trauma-related disorders) and well-being (e.g., psychological well-being), which could position drama as an adjunctive method of mental health care. This original review offered the newer, more comprehensive recommendations for drama-based intervention based on evidence.

## 1. Introduction

The COVID-19 pandemic has severely disrupted people’s lives. In addition to the physical threat, the outbreak has sparked a worldwide mental health crisis. A number of variables, such as fear of infection, treatment ambiguity, financial loss, and social isolation, affected the mental health of the entire population [1,2,3,4,5]. The prevalence of depression and anxiety disorders increased by more than 25% in the first year of the pandemic [6]. Yet, the long-term effects of the COVID-19 pandemic are substantial and will persist for years. After the pandemic, people may experience behavioural changes and emotional anguish [7,8]. Anxiety (ranging from 6.5% to 63%), depression (4% to 31%), post-traumatic stress disorder (12.1% to 46.9%), and cognitive-functional difficulties (4.4% to 17.4%) were observed in persons following coronavirus infection, according to a review of 34 studies [9]. Moreover, current evidence reveals that mental health issues may arise after the pandemic, owing to the environmental stressors, the loss of loved ones, or the change in lifestyle; consequently, early interventions are suggested to improve post-pandemic mental health and support long-term recovery [10,11,12,13].

Mental health is an integral part of an individual’s overall health and well-being. People with better mental health may connect, function, cope, and flourish more effectively [14]. Self-efficacy, independence, and intellectual abilities are regarded as critical components of mental health and well-being [15,16]. Moreover, cognitive and communication abilities may help individuals in adapting to their environment and achieving life satisfaction [17]. Several therapies, such as art, exercise, social skills training, and mindfulness-based interventions, have been demonstrated to enhance positive life experiences, identity, and adaptive functioning [18,19,20,21]. Despite the fact that numerous studies have been undertaken on mental health intervention, there is a need for a diversity of strategies that may support, enable, and sustain mental health improvement innovation.

Drama-based intervention is a creative form of psychotherapy that promotes psychological growth and transformation through the systematic and intentional use of drama and theatre techniques. The concept of role in dramatic context is taken as the central position in various multidisciplinary sources, such as dramatic ritual, which link drama to the treatment phase [22]. By playing a wide repertoire of roles, individuals have learned skills and gained exposure to different perspectives in order to function as both human and social beings [23]. Methods including theatre games, improvisation, psychodrama, storytelling, puppetry, role reversal, playback theatre, and theatre of the oppressed are employed to change the state of illness and crisis, which is the essence of drama-based intervention [24,25,26].

In healthcare settings like clinics, care homes, and community centres, drama appears to support individuals, groups, and families by facilitating communication and challenging negative perspectives [27]. Previous findings highlighted the significance of drama activities in allowing clients to express and tolerate depressive emotions [28,29,30]. In the context of treating trauma-related issues, drama therapy innovations have been shown to benefit the recovery process by assisting patients in developing a positive self-image, confidence, and appreciation of aesthetics [30,31,32]. The effects of drama on people with intellectual and developmental problems have been the subject of multiple investigations at an inpatient psychiatric hospital. Clients’ levels of self-esteem, self-expression, and social skills were found to increase as a result of participating in drama [33,34,35,36]. The COVID-19 lockdown period can be difficult for some people, but a special online drama programme has been supplied to help people deal with issues including loneliness, isolation, and traumatic loss [37,38,39,40].

Preserving mental health and cultivating well-being could be considered as methods for enhancing humanity’s ability to overcome the short- and long-term consequences of the COVID-19 pandemic [41]. In this challenging moment, it is vital to continue documenting drama interventions on mental health issues as an innovative strategy to improve rehabilitation services and establish effective healthcare programmes. Such studies on drama and theatre as treatments would seem significant in promoting well-being. However, as Cheek-O’Donnell [42] points out, drama therapy is currently in the crucial stage of intervention research and empirical investigation. It is imperative that both academics and practitioners have access to evidence-based recommendations about the efficacy of drama-based interventions in improving mental health and well-being.

The quantitative procedures of meta-analysis utilise comparisons of numerical results from a few research studies to examine the usefulness of interventions given in a variety of studies [43]. In this paper, meta-analysis is employed to investigate the effect of drama as an intervention on mental health and well-being. It focuses on studies published after late December 2019, when the outbreak of the coronavirus disease 19 (COVID-19) was reported [44]. The paper explores the productive technique of implementing a drama-based intervention with various populations in order to assess the effect by drawing on the experiences and changes of participants. The aim of the study is to contribute to the systematisation of drama-based treatment and provide evidence-based recommendations for clinicians, drama practitioners, and researchers.

## 2. Materials and Methods

This systematic review and meta-analysis were conducted in accordance with the preferred reporting items for systematic reviews and the meta-analyses (PRISMA) statement [45]. The protocol was registered on the International Platform of Registered Systematic Review and Meta-analysis Protocols (INPLASY). The registration number is INPLASY2022120030.

### 2.1. Search Strategy

Four databases (PubMed, Cochrane, Web of Science, and ScienceDirect) from December 2019 until October 2022 were used to search for potential studies. The following key terms were used: (1) “mental health” OR “psychological wellbeing”; OR (2) “mental health” OR “health, mental” OR “mental hygiene” OR “hygiene, mental” OR “psychological wellbeing” OR “psychological wellness” OR “psychological ill being” OR “ill being psychological” AND (3) “drama” OR “psychodrama” OR “role playing” OR (4) “drama” OR “dramas” OR “drama therapy” OR “therapy, drama” OR “dramatherapy” OR “playing, role” OR “playings, role” OR “role playings” AND (5) “from 2020–2022”. The detailed search strategy is shown in Table 1 (PubMed is used as an example).

### 2.2. Inclusion Criteria

The PICOS (population, intervention, comparison, outcomes, study design) framework has been applied to determine the scope of the research process. (1) Patients in need of mental health and well-being improvement; (2) experimental group with drama-based intervention; (3) control group with no treatment or routine care; (4) outcome indicators such as quality of life, psychological well-being, depression, anxiety, trauma-related disorders, cognitive functioning, communication skills, and self-esteem; and (5) studies with controlled group and pre/posttest that reported the effect of intervention.

### 2.3. Exclusion Criteria

(1) Studies without measurable results or outcomes that were not reported; (2) studies of case reports, vignettes, or conference abstracts; (3) studies with incomplete or unreported data; (4) meta-analysis and review.

### 2.4. Study Selection

All of the literature describing drama intervention or programme was screened. Initially, the titles of the literature were separately checked by two researchers for duplication, non-measurable outcome studies, review papers, conference papers, and protocols. If the article’s title and abstract suggested that a drama intervention or practice was implemented, the full text of the article was downloaded as a potentially eligible study. The entire text was then independently reviewed and identified for inclusion by both researchers. A consensus was reached with the third researcher to resolve any discrepancies.

### 2.5. Data extraction

A pre-selected eight-item data extraction form was used to record data for inclusion in the study under the following headings: (1) author, (2) year of publication, (3) country, (4) population, (5) mean age, (6) sample size, (7) intervention, and (8) outcome measure. The risk of bias in the included studies was utilising a quality instrument validated by The Cochrane Collaboration^®^ [46]. Three levels of bias were assigned to trials: high risk, low risk, and unclear risk [47]. The seven domains were evaluated: (1) random sequence generation, (2) allocation concealment, (3) blinding of the participants and personnel, (4) blinding of outcome assessment, (5) incomplete outcome data, (6) selective reporting, and (7) other sources of bias (the conflict of interests or funding sources). Intervention studies involving humans that require ethical approval must identify the approving authority and the corresponding code.

### 2.6. Data Analysis

The primary outcome was the size of the drama-based intervention’s effect on the experimental group relative to the control group, or the comparison of pre- and post-test. Review Manager (RevMan) version 5.4 was utilised to conduct a meta-analysis of the results’ effects. All of the outcome data in this meta-analysis were continuous and presented as means with standard deviations (SD). Mean differences (SMD) with 95% confidence intervals (CI) were used to figure out a meta-analysis. A random effects model was selected since the effect sizes were pooled [48]. A *p*-value below 0.05 was considered statistically significant. The I^2^ and chi-square tests were used to determine the presence of heterogeneity. Effect size = 0.2 is considered a “small” effect size, 0.5 represents a “medium” effect size, and 0.8 is considered a “large” effect size according to Cohen’s guidelines [49].

## 3. Results

### 3.1. Study Identification and Selection

A total of 452 studies were identified from the electronic database. Due to duplication, 141 articles were removed. Ninety-six irrelevant articles (e.g., unrelated to drama, mental health, and well-being) and fifty-nine articles without abstracts were also removed. The remaining 156 documents were reviewed in abstract, and 87 documents were excluded (for reasons including case reports, conference papers, or reviews). After that, 97 potentially eligible records were retrieved for full text, whereas 28 records were not able to be accessed in full text (e.g., publishers’ restrictions) and were therefore excluded. The remaining 69 studies were read in full text, and 44 documents were again excluded (for reasons including no drama intervention, no quantitative outcome measure, or incomplete data). Finally, 25 documents were included in this study. Figure 1 shows the result of the screening process.

### 3.2. Characteristics of the Included Studies

In total, 25 studies with 797 participants were included. Ten studies were conducted in Europe, eight in the Middle East, four in Asia, and three in the United States. There were 10 distinct forms of drama intervention. The characteristics of the included studies are detailed in Table 2.

#### 3.2.1. Sample Characteristics

Sample sizes ranged from 5 to 114 participants, with age (mean + SD) ranging from 7.08 (1.53) to 82.62 (7.92); the majority of participants were female. Participants with trauma-related depression were included in five studies. Caregivers of children with neurodevelopmental disorders, cerebral palsy, or who were at risk were included in three studies. Two studies on children with attention-deficit/hyperactivity disorder have been conducted. Other studies included participants with social anxiety disorder, community-dwelling, epilepsy, chronic schizophrenia, hearing-impaired adolescents, and dementia patients.

#### 3.2.2. Intervention Characteristics

The primary interventions included were psychodrama (12 studies) and drama therapy (4 studies). Other interventions, such as theatre performance, playback theatre, sociodrama, and theatre of the oppressed, were included. The majority of sessions were conducted once per week over 6–24 weeks, with durations of 1.5–2.5 h. In studies involving older and younger participants, however, shorter time periods (40–60 min) were allocated. In addition, a six-month study was conducted every two months for a duration of four days. Most of the interventions were conducted by occupational therapists, whereas instructors for the inpatient programme were clinical psychiatrists or specialists.

#### 3.2.3. Outcome Measures

The outcome measures consisted of eight mental-health-related components that were administered as follows: quality of life in nine comparisons, psychological well-being in five comparisons, depression in seven comparisons, anxiety in three comparisons, trauma-related disorders in seven comparisons, cognitive functioning in four comparisons, communication skill in four comparisons, and self-regard in five comparisons. Regarding study design, twelve studies were accurately described as controlled trials, including five randomised controlled trials (RCTs), while thirteen were experimental studies with pre- and post-test groups.

### 3.3. Risk of Bias Analysis

Risks of bias were judged based on the Cochrane guidance. Regarding random sequence generation, four studies were deemed high-risk due to the absence of randomization procedures [50,52,65,70]. In terms of allocation concealment, three studies were judged high-risk reported no use of concealment in the allocation procedure [65,66,70]. Two studies [52,66] were evaluated as high risk regarding participants and personnel blinding. Meanwhile, the lack of blinding in outcome assessment led to a high risk classification for one study [50]. Except for four studies where the information was not reported [51,60,62,64], all studies were evaluated as low risk for incomplete outcome data due to the low dropout rate throughout the intervention. Regarding selective reporting, seventeen studies were rated as low risk, and the remaining eight studies were judged to be unclear. Concerning other biases, three studies did not identify the conflict of interest [56,58,72], one reported the author received research honoraria [52], and one indicated the authors were on the board of an entity supporting the research [69]. Figure 2 and Figure 3 display specific features.

### 3.4. Meta-Analysis

All effects and heterogeneities for quality of life, psychological well-being, depression, anxiety, trauma-related disorders, communication skills, cognitive functioning, and self-regard were tested. The results are presented in Table 3.

#### 3.4.1. Effect on Quality of Life

A total of nine studies involving 213 participants provided the results of the meta-analysis regarding quality of life. On the basis of comparisons, significant difference in the use of drama-based intervention on quality of life was observed, with a considerable impact size (SMD = 1.26, 95% CI = 0.33 to 2.20, *p* = 0.008). However, the level of heterogeneity was substantial (I^2^ = 91).

#### 3.4.2. Effect on Psychological Well-Being

Psychological well-being was measured in five investigations with a total of 208 individuals. There was no statistically significant difference, and great heterogeneity (I^2^ = 96) was displayed amongst studies. The effect size of the drama-based intervention was deemed to be substantial (SMD = 1.40, 95% CI −0.32 to 3.12, *p* = 0.11).

#### 3.4.3. Effect on Depression

In seven investigations with 225 participants, the effect of a drama-based intervention for depression was evaluated. In the comparison of three controlled trials and four pre/post-test studies, significant differences were shown in favour of drama-based intervention in depression assessments (SMD = 0.70, 95% CI −0.03 to 1.42, *p* = 0.03). The I^2^ value in this meta-analysis, however, was high (I^2^ = 85).

#### 3.4.4. Effect on Anxiety

Assessments of anxiety were included in three studies with a total sample size of 99. There were two controlled trials and one pre/post-test study in the comparison, revealing positive effect of drama-based intervention on anxiety but no statistically significant difference (SMD = 1.10, 95% CI −0.24 to 2.45, *p* = 0.11), with a high degree of heterogeneity (I^2^ = 87) across studies.

#### 3.4.5. Effect on Trauma-Related Disorders

To evaluate trauma-related disorders, four studies with 169 people were analysed. The comparison included one controlled trial and three pre/post-test investigations. Meta-analysis revealed a positive effect of drama-based intervention on trauma-related disorders (SMD = 0.70, 95% CI 0.23 to 1.17, *p* = 0.003), with a substantial heterogeneity (I^2^ = 66).

#### 3.4.6. Effect on Communication Skills

In four trials with a total of 86 participants, communication skills were assessed. The comparison included three controlled trials and one pre/post-test study. Meta-analysis found that drama-based intervention had a great impact on enhancing communication skills (SMD = 1.76, 95% CI −0.06 to 3.57), while there was no statistically significant difference (*p* = 0.06), and the I^2^ value was high (I^2^ = 91).

#### 3.4.7. Effect on Cognitive Functioning

Seven studies with 189 participants were used to measure cognitive functioning. There were three controlled trials and four pre/post-test studies in the comparison. Studies applying drama-based intervention on cognitive performance found the large impact size, with statistically significant difference (SMD = 2.50, 95% CI 0.77 to 4.23, *p* = 0.005). However, the level of heterogeneity was substantial in this meta-analysis (I^2^ = 96).

#### 3.4.8. Effect on Self-Regard

Self-regard was measured using data from five studies with 215 participants. In the comparison, there were two controlled trials and three pre/post-test investigations. Meta-analysis revealed that drama-based intervention had no statistically significant effect on self-esteem (SMD = 1.40, 95% CI −0.06 to 2.86, *p* = 0.06), and the I^2^ value was high (I^2^ = 95).

#### 3.4.9. Overall Effect with Controlled Groups

Drama-based intervention had a positive effect with controlled groups on cognitive functioning (SMD = 1.58, 95% CI 0.62 to 2.54, *p* = 0.001). No statistically significant difference was observed between controlled groups on quality of life (SMD = 2.08, 95% CI −0.33 to 4.49, *p* = 0.09), psychological well-being (SMD = 1.69, 95% CI −0.45 to 3.83, *p* = 0.12), depression (SMD = 1.77, 95% CI −0.35 to 2.70, *p* = 0.13), trauma-related disorders (SMD = 0.14, 95% CI −0.46 to 0.74, *p* = 0.65), anxiety (SMD = 0.88, 95% CI −0.82 to 2.58, *p* = 0.31), communication skills (SMD = 1.11, 95% CI −0.68 to 2.90, *p* = 0.06), and self-regard (SMD = 2.83, 95% CI −0.90 to 6.56, *p* = 0.14). The largest positive effect size was self-regard, followed by quality of life, psychological well-being, depression, communication skills, cognitive functioning, anxiety, and trauma-related disorders. See Figure 4.

#### 3.4.10. Overall Effect with Pre/Posttest Groups

Drama-based intervention with pre/post-test groups was effective to reduce depression (SMD = 0.42, 95% CI 0.05 to 0.78, *p* = 0.03), anxiety (SMD = 1.74, 95% CI 0.16 to 3.32, *p* = 0.03), and trauma-related disorder (SMD = 0.90, 95% CI 0.52 to 1.28, *p* < 0.0001). It is also effective to improve quality of life (SMD = 0.86, 95% CI 0.06 to 1.67, *p* = 0.04), communication skills (SMD = 4.98, 95% CI 1.91 to 8.04, *p* = 0.001), and self-regard (SMD = 0.39, 95% CI 0.14 to 0.65, *p* = 0.002). No statistically significant difference was observed between pre/post-test groups on psychological well-being (SMD = 0.46, 95% CI −0.69 to 1.62, *p* = 0.43) and cognitive functioning (SMD = 3.47, 95% CI −1.02 to 7.97, *p* = 0.13). The largest effect size was communication skills, followed by cognitive functioning, anxiety, trauma-related disorder, quality of life, psychological well-being, depression and self-regard. See Figure 5.

### 3.5. Evaluation of Publication Bias

The symmetry of the funnel plot was utilised to determine the publication bias of the meta-analysis results [75].The funnel plot of standard errors by SMD was assessed according to its symmetry, and the results are presented in Figure 6. The dotted line on each side of the figure represent the 95% confidence intervals. The middle line represents the effect of the meta-analysis as a whole. No apparent publication bias among the studies on quality of life, depression, anxiety, psychological well-being, cognitive functioning, trauma-related disorder, communication skills, and self-regard was observed, as indicated by visual observation of the funnel plots.

## 4. Discussion

This systematic review and meta-analysis evaluated studies that employed drama-based interventions to promote mental health and well-being in the COVID-19 and post-pandemic periods. A total of 25 studies representing 797 participants were identified. Overall, drama-based interventions were shown to have the most consistent favourable effect on trauma-related disorders, cognitive functioning, quality of life, and depression. Moreover, the study found that drama may be considerably beneficial but not significant in increasing psychological well-being and communication skills. Regarding the duration of drama-based intervention, except for two studies that employed 64 [55] and 72 weeks [56], the majority of the publications indicated that drama as a supplemental treatment was viable and acceptable with durations of 8–12 weeks.

In terms of forms and techniques of drama-based intervention, psychodrama (12 studies) and drama therapy (4 studies) were the most prevalent ones included. Other forms like theatre performance, playback theatre, theatre of the oppressed, and sociodrama were also included. The included studies found that the psychodrama had a statistically significant beneficial effect on lowering anxiety levels [50,62,74] and enhancing communication skills [51,64]. The playback theatre had the greatest effect on psychological well-being [58]. In addition, drama therapy was more effective than other cognitive functioning programmes applied in selected studies [54,61]. Furthermore, the findings revealed a positive psychological effect of organised short-term playback theatre involvement in community-dwelling older persons, indicating that the community drama may provide the elderly with an opportunity to enhance their existing well-being [58].

Initial results from the meta-analysis showed that drama-based interventions were effective in reducing symptoms of trauma-related disorders, according to the outcomes of four included studies. This is consistent with the finding of Yu et al. [74], who discovered that antidepressants combined with psychodrama were more effective at enhancing the coping style of patients with childhood traumatic depression than combined with a general health education intervention, thus providing a new clinical intervention option. In addition, Miguel and Pino-Juste [63] demonstrated that the psychodrama method (warming-up, action, and sharing) had a positive effect on domestic violence victims. Other two included studies added to the evidence that psychodrama is beneficial for reducing PTSD in inpatient substance abuse treatment patients [55,56]. It might be due to the fact that drama is more likely to assist individuals in expressing their difficulties, discovering their conflicts, and then confronting them [76,77]. The advantages were underlined in this review.

Meanwhile, the meta-analysis of seven studies discovered a statistically significant and favourable effect of drama-based interventions on cognitive functioning. For instance, the psychodrama technique may assist youngsters with attention deficit/hyperactivity disorder (ADHD) to reduce their aggressive behaviour and concentration difficulties [62]. Furthermore, drama therapy can be utilised as an effective intervention to lower the expenses of ADHD treatment, particularly for strengthening working memory in adolescents with ADHD [59]. Using a dramatic diary and monologue, cognitive-behavioral psychodrama group therapy promoted critical thinking and decreased defensiveness in inadequately guarded male adolescents [53]. This may be because of the way that drama encourages individuals to express repressed tensions and emotions in a safe environment by fostering spontaneity, inventiveness, and rational reasoning [54,78].

Moreover, nine studies included in the review demonstrated that drama-based intervention improved the quality of life. According with the findings of meta-analysis, the results of Vlotinou et al. [73] implied that drama activities (e.g., emotional expression, role-playing) may improve life quality of people with epilepsy by addressing their fear and loss of control. Besides, Simsek et al. [68] showed that quality of life of caregivers of children with cerebral palsy can be enhanced by increasing hope and self-confidence through warm-up, action, and sharing stages in psychodrama. Further research added to the findings that drama intervention fostered more favourable views towards the illness and social environment by gaining empathy and allowing participants to perceive themselves in diverse roles [63,67].

Additionally, the review confirmed the significance of drama-based intervention in the treatment of depression. In a meta-analysis, the statistical significance of seven investigations was determined. For instance, the included study by Sevi et al. [67] indicated that psychodrama sessions (e.g., dramatic games) helped alleviate depressive symptoms in patients with chronic schizophrenia by boosting sharing, group interaction, and a sense of belonging. Besides, the finding of Keisari et al. [58] showed that playback theatre, which integrates dramatic expression with the exploration of life stories in a group creative process, had a positive psychological effect in community-dwelling older adults with depressive psychological distress. Several studies have added to the evidence that drama improves mental health and reduces depressive symptoms through self-reflection and personal development. The drama programme provided therapeutic benefit and acted as a vehicle for the participants’ positive transformation [50,52,60].

It is also noted that the results suggested that drama-based intervention was beneficial but not statistically significant for enhancing psychological well-being. For example, a psychodrama programme improved the psychological balance of adolescents who had experienced a traumatic incident; the correlations showed increased psychological progress [71]. Furthermore, the results may point to the potential role of drama-based intervention in improving communication skills. This is consistent with the findings of Jang et al. [57], who discovered that the sociodrama programme improved parent–child communication for mothers of children with neurodevelopmental disorders. In accordance with additional findings [51,64,65], participants gained abilities via interpersonal interactions in the dramatic activities.

The review consists of twelve controlled groups, including five randomised controlled trials (RCTs), and thirteen experimental studies with pre- and post-testing. Depression, anxiety, trauma-related disorder, quality of life, communication skills, and self-regard were all positively affected by a drama-based intervention supported by pre/post-testing, whereas the overall effect of intervention with controlled trials had only positive effects on cognitive functioning. The current findings revealed that controlled groups were insufficient to demonstrate the efficacy of drama interventions on mental health and well-being. It suggested that additional well-designed controlled trials comparing experimental and control groups are required to evaluate the impact of drama-based interventions. More RCTs, particularly those with high-quality designs, were also called to provide causal explanations for the difference between pre- and post-values.

Several studies included in this review suggested drama-based intervention was feasible to cope with the COVID-19 pandemic. Consistent with the findings, Giacomucci et al. [56] discovered that people with active trauma from COVID-19 who participated in drama sessions reported a reduction in depression and PTSD symptoms. Notably, the COVID-19 social distancing decreased social contact engagement and may be risk factors for isolation, anxiety, and depression [79]. A number of included studies have provided solutions that allowed social connectedness through drama activities and generated the stimulating senses of connection and empathy in others [52,58,67], thereby enabling individuals to address issues of shared concern and increased their ability to embrace challenge in the pandemic and post-pandemic period. During quarantine, Cheung et al. [52] provided people with severe mental illness with online drama intervention via Zoom, which was considered a feasible strategy in the setting of the epidemic. This study constructed a convincing case for the usefulness of drama intervention in the digital space and proposed a novel strategy for dealing with pandemic realities.

## 5. Strengths and Limitations

In this study, the techniques of systematic review and meta-analysis were used to analyse the effects of drama on mental health and well-being of various populations during the COVID-19 and post-pandemic eras. This is the first review and meta-analysis to critically examine the evidence for the use of drama in mental health care. The findings of the selected studies provided crucial evidence of the effectiveness of drama-based intervention on mental health issues. This original review on the treatment of drama throughout individuals who required mental health care by different drama programmes provided more recent and comprehensive evidence-based recommendations.

The limitations of the study should be acknowledged. First, the literature search was restricted to the publications of the COVID-19 and the post-pandemic period (from December 2019 to October 2022); thus, some theoretically significant earlier contributions may have been overlooked. The number of included papers was modest, and several of the studies had small sample sizes, which may have compromised the reliability of the meta-analytic conclusions. Besides, a stratified analysis was not possible due to the extensive range of participant characteristics. Moreover, several studies in the present evidence base were conducted in quasi-experiment groups, which were inadequate to advance knowledge of the effects of drama-based intervention. Furthermore, despite the researchers’ best attempts, heterogeneity between studies could not be avoided.

## 6. Conclusions

This review and meta-analysis concludes that drama has the potential to improve mental health (e.g., trauma-related disorders) and well-being (e.g., psychological well-being), positioning it as a supplement to mental health care during and after the COVID-19 pandemic. Drama-based intervention is increasingly offered in healthcare settings as part of a variety of complementary therapies. Future research may examine the effects of drama-based interventions on individuals with post-COVID-19 pandemic trauma to better comprehend the correlations between drama activity and therapeutic effect. To further understand how drama interventions may be utilised as psychological prescriptions, it is recommended that future study compare drama intervention to other therapeutic treatments and/or compare different forms of drama programme. Moreover, telehealth and other technological advances may help improve the efficacy of drama intervention, which might be studied in the future.

## Figures and Tables

**Figure 1 healthcare-11-00839-f001:**
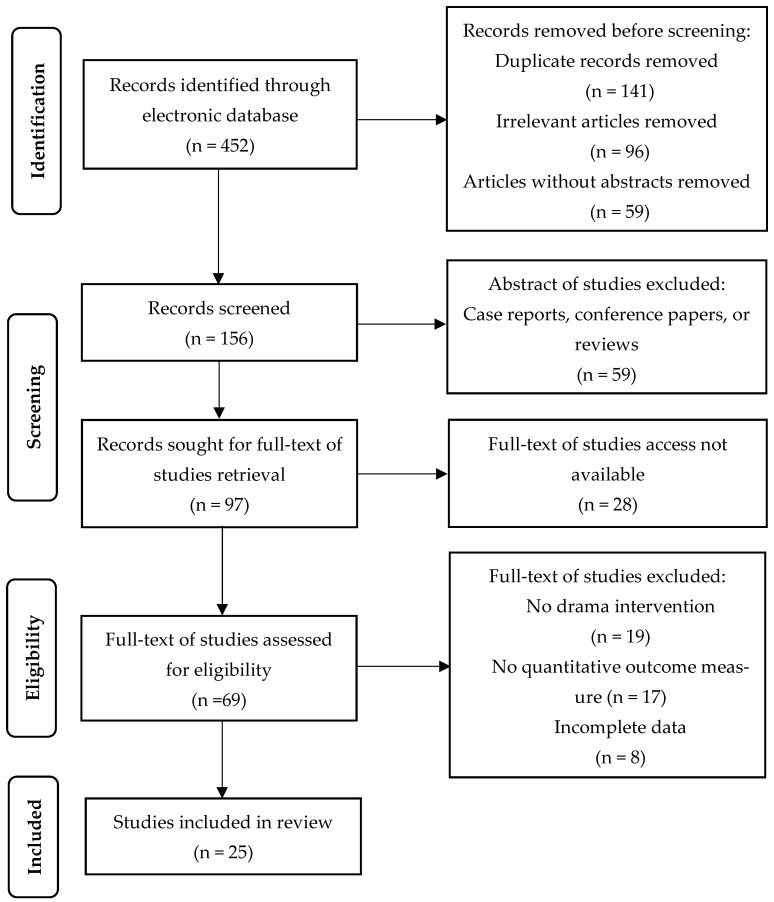
Flow diagram of literature selection.

**Figure 2 healthcare-11-00839-f002:**
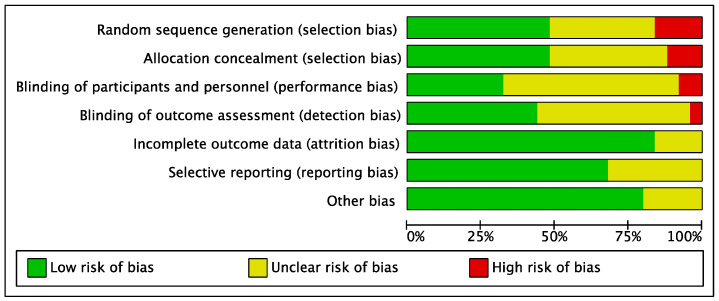
Risk of bias graph for included studies.

**Figure 3 healthcare-11-00839-f003:**
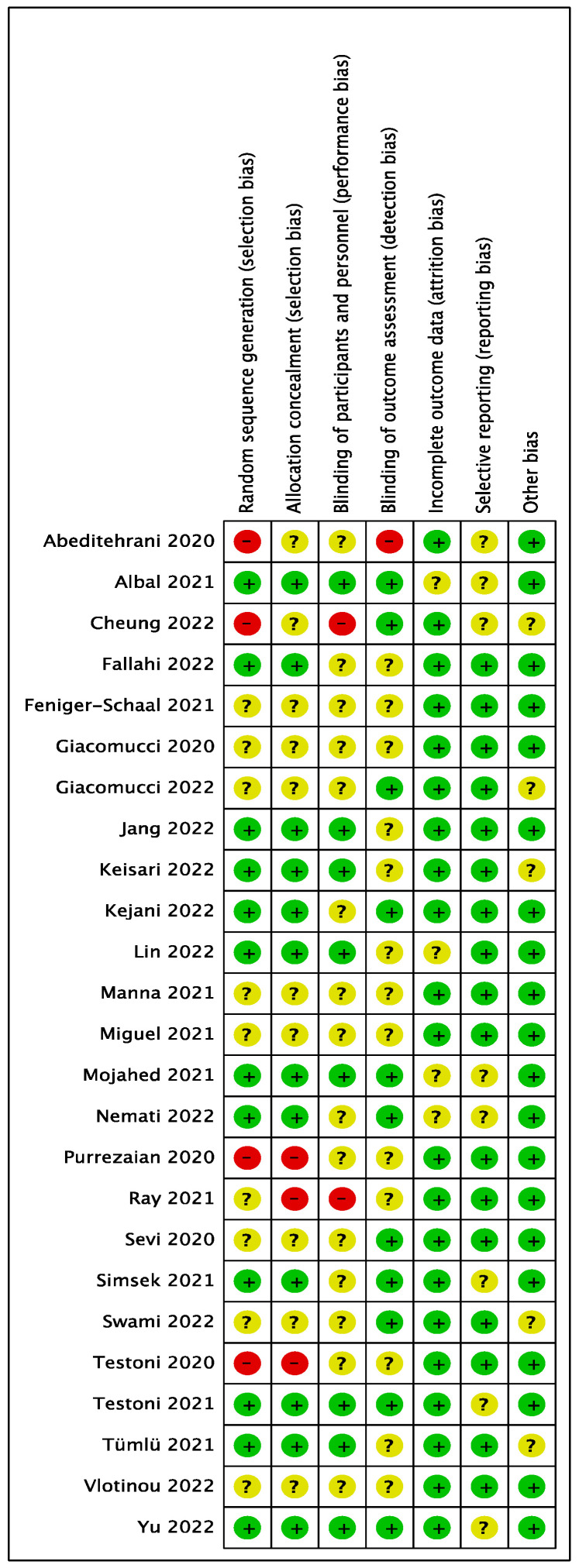
Risk of bias summary for each included study [50,51,52,53,54,55,56,57,58,59,60,61,62,63,64,65,66,67,68,69,70,71,72,73,74].

**Figure 4 healthcare-11-00839-f004:**
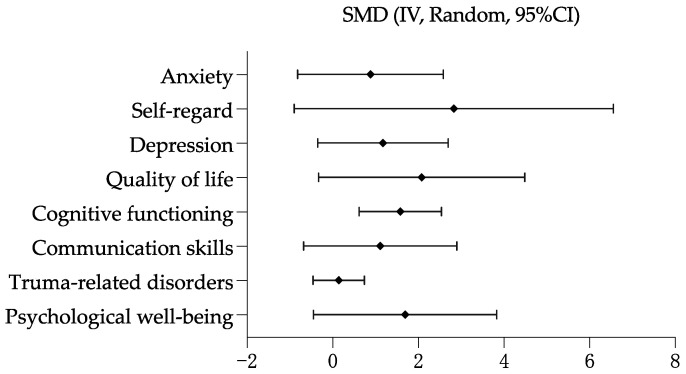
Overall effectiveness of drama-based intervention with controlled groups.

**Figure 5 healthcare-11-00839-f005:**
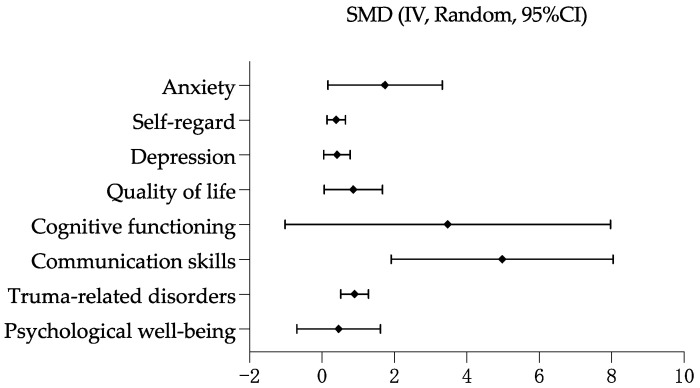
Overall effectiveness of drama-based intervention with pre/post-test groups.

**Figure 6 healthcare-11-00839-f006:**
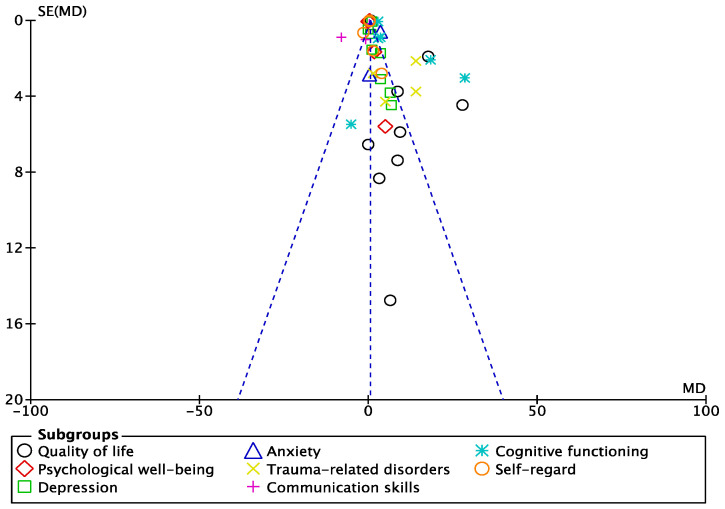
Funnel plots for visual assessment of the presence of publication bias.

**Table 1 healthcare-11-00839-t001:** Search strategy on PubMed.

#1	“mental health”[MeSH] OR “psychological wellbeing”[MeSH]
#2	(((((((Mental Health [Title/Abstract]) OR Health, Mental [Title/Abstract]) OR Mental Hygiene [Title/Abstract]) OR Hygiene, Mental [Title/Abstract]) OR psychological wellbeing [Title/Abstract]) OR psychological wellness [Title/Abstract]) OR psychological ill being [Title/Abstract]) OR ill being psychological [Title/Abstract]
#3	#1 OR #2
#4	“drama”[MeSH] OR “psychodrama”[MeSH] OR “role playing”[MeSH]
#5	(((((((drama [Title/Abstract]) OR dramas [Title/Abstract]) OR drama therapy [Title/Abstract]) OR therapy, drama [Title/Abstract]) OR dramatherapy [Title/Abstract]) OR playing, role [Title/Abstract]) OR playings, role [Title/Abstract]) OR role playings [Title/Abstract]
#6	#4 OR #5
#7	“from 2019–2022”
#8	#3 AND #6 AND #7

**Table 2 healthcare-11-00839-t002:** Characteristics of the studies included in the meta-analysis.

First Author (Year)	Country	Population	Design	Age (Mean + SD)	Total/Male/Female	Intervention	Outcome Measure
Abeditehrani (2020) [50]	The Netherlands	Adult female patients with SAD	Pre/post treatment	T: 36.6 (17.8) C: NA	T: 5/0/5 C: NA	Psychodrama Period: 12 weeks Freq: once a week Duration: 2.5 h	Quality of life: QLS; Depression: BDI; Anxiety: LSAS
Albal (2021) [51]	Turkey	Psychiatric nurses	Randomised controlled trial	T: 35.54 (9.03) C: 39.62 (5.12)	T: 13/1/12 C: 13/0/13	Psychodrama Period: 8 weeks Freq: once a week Duration: 2 h	Communication skills: CSI
Cheung (2022) [52]	USA	People with SMI	Pre/post treatment	T: 51.5 (9.4) C: NA	T: 8/4/4 C: NA	Drama therapy Period: 12 weeks Freq: once a week for 10 weeks Duration: 1.5 h	Psychological well-being: BPRS; Quality of Life: QLESS; Depression: PSS
Fallahi (2022) [53]	Iran	Inadequate guardian male adolescents	Controlled trial with pre/posttest	T: NA C: NA	T: 15/15/0 C: 15/15/0	Psychodrama Period: 6 weeks Freq: NA Duration: 2 h	Cognitive functioning: EMI
Feniger-Schaal (2021) [54]	Israel	Mothers of children-at-risk	Pre/post treatment	T: 37 (5.9) C: NA	T: 40/0/40 C: NA	Drama therapy Period: 10 weeks Freq: NA Duration: 1.5 h	Cognitive functioning: CBCL
Giacomucci (2020) [55]	USA	People with PTSD	Pre/post treatment	T: 41.34 (12.53) C: NA	T: 86/40/44/ 2 transgenders C: NA	Psychodrama Period: 64 weeks Freq: twice a week Duration: 2.25 h	Trauma-related disorders: PCL
Giacomucci (2022) [56]	USA	People with depression and PTSD	Pre/post treatment	T:40.60 (11.79) C: NA	T: 20/6/14 C: NA	Psychodrama Period: 72 weeks Freq: twice a week Duration: 2.25 h	Trauma-related disorders: PCL
Jang (2022) [57]	South Korea	Mothers of children with ND	Non-randomised controlled experiment	T: 42.62 (6.29) C: 42.67 (7.34)	T: 16/0/16 C: 18/0/16	Sociodrama Period: 6 weeks Freq: once a week Duration: 2.5 h	Commination skills: PACS
Keisari (2022) [58]	Israel	Community-dwelling older adults	Randomised controlled trial	T: 78.65 (6.91) C: 80.60 (6.81)	T: 40/9/31 C: 38/6/32	Playback theatre Period: 12-week Freq: once a week Duration: 1.5 h	Quality of life: SLS; Depression: GDS; Self-regard: SEC
Kejani (2020) [59]	Iran	ADHD primary school children	Quasi-experiment with pre/posttest	T: NA C: NA	T: 21/10/11 C: 24/12/12	Drama therapy Period: 6 weeks Freq: twice a week Duration: 1.5 h	Cognitive functioning: WISC
Lin (2022) [60]	Taiwan	Patients with dementia	Randomized controlled trial with pre/posttest	T: 82.62 (7.92) C: 82.58 (7.74)	T: 23/4/19 C: 19/5/14	Drama therapy Period: 8 weeks Freq: once a week Duration: 1.5 h	Quality of life: ADLS; Psychological well-being: MMSE; Depression: CSDD
Manna (2021) [61]	India	Children with ASD	Pre/post treatment	T: NA C: NA	T: 16/11/5 C: NA	Drama therapy Period: NA Freq: NA Duration: NA	Cognitive functioning: CSCR
Mojahed (2021) [62]	Iran	Children with ADHD	Randomised controlled trial with pre/posttest	T: 9.92 (1.381) C: 9.79 (1.285)	T: 24/24/0 C: 24/24/0	Psychodrama Period: 10 weeks Freq: once a week Duration: 2 h	Cognitive functioning: CBCL; Anxiety: SCAS
Miguel (2021) [63]	Spain	Women victims of intimate partner violence	Pre/post treatment	T: 49 (NA) C: NA	T: 17/0/17 C: NA	Drama therapy, theatre of the oppressed, and psychodrama Period: 20 sessions Freq: NA Duration: 2 h	Trauma-related disorders: SPSS; Quality of life: QLS; Depression: BDI; Self-regard: SES
Nemati (2022) [64]	Iran	Adolescents with hearing loss	Quasi-experiment with pre/posttest	T: 13.9(1.46) C: 14.3 (0.86)	T:12/ NA/ NA C: 12/ NA/ NA	Psychodrama Period: 5 weeks Freq: twice a week Duration: 1.5 h	Communication skills: QCST
Purrezaian (2020) [65]	Iran	Hospitalised children with cancer	Pre/post treatment	T: 11(1.58) C: NA	T: 5/2/3 C: NA	Psycho-art-drama Period: 8 sessions Freq: NA Duration: 40–60 min	Communication skills: BPSEIH
Ray (2021) [66]	Israel	Traumatised adults	Pre/post treatment	T: NA C: NA	T: 10/1/9 C: NA	Autobiographical therapeutic performance Period: 10 month Freq: NA Duration: NA	Cognitive functioning: BRIEF-A
Sevi (2020) [67]	Turkey	Patients with chronic schizophrenia	Pre/post treatment	T: 55.52 (7.45) C: NA	T: 31/19/12 C: NA	Psychodrama Period: 19 session Freq: once a week Duration: 1.5−2 h	Depression: CDS; Quality of life: QLS
Simsek (2021) [68]	Turkey	Mothers of children with cerebral palsy	Controlled trial	T: 30.8 (7.0) C: 33.1 (7.4)	T: 8/0/8 C: 14/0/14	Psychodrama Period: 8 weeks Freq: once a week Duration: 2 h	Quality of life: QLS
Swami (2022) [69]	UK	Children between 5 and 9 years	Pre/post treatment	T: 7.08 (1.53) C: NA	T: 99/45/54 C: NA	Theatrical performance Period: 8 weeks Freq: NA Duration: NA	Self-regard: BAS
Testoni (2020) [70]	Italy	Prisoner with substance dependence	Pre/post treatment	T: 34 (8.71) C: NA	T: 7/7/0 C: NA	Psychodrama Period: 24 weeks Freq: once a week Duration: 1.5 to 2 h	Cognitive functioning: SAI-R; Self-regard: GSE;
Testoni (2021) [71]	Italy	Traumatic high school students	Controlled trial with pre/posttest	T: 15.98 (1.12) C: 16.14 (1.00)	T: 45/18/27 C: 37/4/33	Psychodrama Period: 5 weeks Freq: NA Duration: 2 h	Psychological well-being: PWS;
Tümlü (2021) [72]	Turkey	Research assistants	Quasi-experiment with pre/posttest	T: 30.9 (3.3) C: 31.6 (2.9)	T: 7/NA/NA C: 7/NA/NA	Psychodrama Period: 10 weeks Freq: once a week Duration: 3 h	Self-regard: SCS
Vlotinou (2022) [73]	Greece	Patients with epilepsy	Pre/post treatment	T: 32.27 (13.55) C: NA	T:15/6/9 C: NA	Occupational therapy program with drama activities Period: 12 weeks Freq: once a week Duration: 2 h	Quality of life: QLS
Yu (2022) [74]	China	Patients with childhood trauma-associated MDD	Randomised controlled trial	T: 25.97 (7.189) C: 28.12 (6.214)	T: 29/7/22 C: 17/2/15	Psychodrama Period: 24 weeks Freq: once eight weeks Duration: 4 days	Depression: BDI; Anxiety: BAI

Note. T: experimental group; C: control group; NA: unavailable; Freq: frequency; LoI: length of intervention; SAD: social anxiety disorder; SMI: serious mental illness; PTSD: post-traumatic stress disorder; ND: neurodevelopmental disorders; ADHD: attention-deficit/hyperactivity disorder; ASD: autism spectrum disorder; MDD: major depressive disorders. QLS: Quality of Life Scale; QLESS: Quality of Life Enjoyment and Satisfaction Scale; ADLS: Activity of Daily Living Scale; SLS: Satisfaction with Life Scale; PWS: Psychological Well-being Scale; MMSE: Mini-mental State Examination; BPRS: Brief Psychiatric Rating Scale; CSDD: Cornell Scale for Depression in Dementia; BDI: Beck Depression Inventory; GDS: Geriatric Depression Scale; PSS: Perceived Stress Scale; CDS: Calgary Depression Scale; SCAS: Spence Children’s Anxiety Scale; BAI: Beck Anxiety Inventory; LSAS: Liebowitz Social Anxiety Scale; CTQ: Childhood Trauma Questionnaire; SPSS: Severity of PTSD Symptoms Scale; PCL: PTSD Checklist; CBCL: Child Behaviour Checklist; WISC: Wechsler Intelligence Scale for Children; EMI: Ricketts’ Engagement, Maturity, and Innovativeness; SAI-R: Revised Spontaneity Assessment Inventory; CSCR: Child’s Skill Scale Rating; BRIEF-A: Behaviour Rating Inventory of Executive Function-Adult version; CBCL: Child Behaviour Checklist; QCST: Queendom Communication Skills Test; CSI: Communication Skills Inventory; BPSEIH: Bio-psycho-social Expressions of Incompatibility in Hospital; PACS: Parent-adolescent Communication Scale; SCS: Self-Compassion Scale; SEC: Self-esteem Scale; GSES: General Self-efficacy Scale; BAS: Body Appreciation Scale.

**Table 3 healthcare-11-00839-t003:** Effects and heterogeneity for comparisons between studies.

			Effects		Heterogeneity	
Outcome	Study Design	SAMPLE SIZE	SMD (IV, Random, 95%CI)	*p*	I^2^ (%)	*p*
Quality of life	Controlled group (*n* = 4)	139	2.08 [−0.33, 4.49]	0.09	96	<0.00001
	Pre/posttest group (*n* = 5)	74	0.86 [0.06, 1.67]	0.04	79	0.0002
	Total (*n* = 9)	213	1.26 [0.33, 2.20]	0.008	91	<0.00001
Psychological well-being	Controlled group (*n* = 4)	202	1.69 [−0.45, 3.83]	0.12	97	<0.00001
	Pre/posttest group (*n* = 1)	6	0.46 [−0.69, 1.62]	0.43	-	-
	Total (*n* = 5)	208	1.40 [−0.32, 3.12]	0.11	96	<0.00001
Depression	Controlled group (*n* = 3)	166	1.17 [−0.35, 2.70]	0.13	95	<0.00001
	Pre/posttest group (*n* = 4)	59	0.42 [0.05, 0.78]	0.03	0	0.82
	Total (*n* = 7)	225	0.70 [−0.03, 1.42]	0.03	85	<0.00001
Anxiety	Controlled group (*n* = 2)	94	0.88 [−0.82, 2.58]	0.31	93	0.0002
	Pre/posttest group (*n* = 1)	5	1.74 [0.16, 3.32]	0.03	-	-
	Total (*n* = 3)	99	1.10 [−0.24, 2.45]	0.11	87	0.0004
Trauma-related disorders	Controlled group (*n* = 1)	46	0.14 [−0.46, 0.74]	0.65	-	-
	Pre/posttest group (*n* = 3)	123	0.90 [0.52, 1.28]	<0.00001	36	0.21
	Total (*n* = 4)	169	0.70 [0.23, 1.17]	0.003	66	0.03
Communication skills	Controlled group (*n* = 3)	86	1.11 [−0.68, 2.90]	0.22	92	<0.00001
	Pre/posttest group (*n* = 1)	5	4.98 [1.91, 8.04]	0.001	-	-
	Total (*n* = 4)	91	1.76 [−0.06, 3.57]	0.06	91	<0.00001
Cognitive functioning	Controlled group (*n* = 3)	123	1.58 [0.62, 2.54]	0.001	81	<0.00001
	Pre/posttest group (*n* = 4)	66	3.47 [−1.02, 7.97]	0.13	98	0.006
	Total (*n* = 7)	189	2.50 [0.77, 4.23]	0.005	96	<0.00001
Self-regard	Controlled group (*n* = 2)	92	2.83 [−0.90, 6.56]	0.14	96	<0.00001
	Pre/posttest group (*n* = 3)	123	0.39 [0.14, 0.65]	0.002	0	0.83
	Total (*n* = 5)	215	1.40 [−0.06, 2.86]	0.06	95	<0.00001

## Data Availability

The data that support the findings of the study are available from the corresponding author, upon reasonable request.

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
