# Peer review of "Effectiveness of Drama-Based Intervention in Improving Mental Health and Well-Being: A Systematic Review and Meta-Analysis during the COVID-19 Pandemic and Post-Pandemic Period"

_healthcare, 2023, doi:10.3390/healthcare11060839_

Round 1

Reviewer 1 Report

I find the article interesting for the scientific community. The analysis of the effectiveness of the drama on mental health and well-being is a topic that allows to study the possible palliative therapies that the drama could consider. For this reason, it is convenient to highlight that some studies highlight that during four weeks of drama instruction, well-being and cognitive functioning (word reading and problem solving) significantly improved among a sample of older adults compared to individuals in a control condition without treatment.

In relation to data extraction, the selection of the eight items to record the data in the study seems interesting to us (1) author, (2) year of publication, (3) country, 142 (4) population, (5) mean age, (6) sample size, (7) intervention, and (8) outcome measure.

The authors have to improve the presentation and clarity of the following figures:

• 3.3. Risk of Bias Analysis

• Figure 3. Risk of bias summary for each included study.

• Figure 5. Overall effectiveness of drama-based intervention with pre/posttest groups.

• Figure 6. Funnel plots for visual assessment of the presence of publication bias.

Section 6. Conclusion should be expanded

Reviewer 2 Report

This is an interesting paper, providing a systematic review and meta-analysis of drama-based interventions conducted during the COVID-19 pandemic and post-pandemic period, aiming to synthesize evidence regarding their effectiveness on a number of well-being and mental health indicators. The study is well-conducted and the paper is well-written, reaching significant findings and conclusions. I therefore think that it deserves publication in the journal, on the condition of important revisions made. The revisions I suggest below concern mainly clarification/correction of terms and processes, and most importantly highlighting the contribution of the study in the discussion.

I would like to start with a disclaimer: I am not an expert in the field of drama-based interventions and I am not familiar with meta-analytic procedures. I therefore cannot comment on the relevance of the literature drawn upon and the quality of contributions regarding drama-based interventions specifically. Neither can I comment on the specificities and the correctness of the analyses conducted.

Below, I outline the areas that I think are in need of revision:

In 2.5. remove “Rick of bias in the included studies”

In 3.1. clarifications are needed regarding the study selection process. Specifically, what do you mean that records were illegible by automation tools? 59 records were removed for other reasons. What were these reasons? 28 records were not retrieved. Why?

In 3.2. you classify the UK separately from Europe. The UK is part of Europe, outside the European Union. You should either have the European Union and UK as separate or leave Europe and include the UK in it. (I would personally go for the latter option).

I do not understand what “Cognitive behaviour” as outcome measure means. Either explain or, better, use a different term.

You seem to arrange the numbering of the studies included in the review based on their appearance in section 3.3. I would re-arrange them according to their appearance in Table 2. At least, there the studies’ number should appear in the Table, to assist the reader in finding them. Finally, some references seem to be missing, e.g. Pinelopi, 2022 (which by the way is a first name, not a surname). This can also be seen from the fact that the studies included are numbered starting from 50 and should go until 75, given that there are 25. However, they go up to 72. Please correct.

There seems to be a significant difference in findings of effectiveness between studies with controlled groups (3.4.9) and studies with pre/post-test groups (3.4.10). Can you provide possible explanations for this in the discussion?

The Discussion needs reorganizing and re-writing in order to draw out clearly the findings and implications of the study. It is right that you start with a summary of the findings regarding effectiveness of drama-based interventions for different wellbeing and mental health indicators, specifying also specific effects of specific drama-based interventions. It is a good idea that you give qualitative information regarding how drama-based interventions affect the different indicators, giving examples from the included studies. You need to change the phrasing there, to make clear that you are drawing upon included studies to give examples, as I was not sure for a while whether you were referring to the studies or making links with other relevant literature. Why do you not include a discussion of quality of life and depression, although they were found relevant? After discussing the effectiveness of drama-based interventions and how they achieve these effects through examples, I think that you should make a case for the feasibility of drama-based interventions, especially in crisis situations, such as the COVID-19 pandemic, for addressing well-being and mental health of the population, by discussing the characteristics of the interventions as found in the studies, i.e. population, duration, problems addressed etc. I think that this would strengthen your argument and lead to recommendations.

Finally, proof reading by a native speaker of the whole paper is required to iron out a few instances of awkward expression in English.

Reviewer 3 Report

This article presented a meta-analytic summary of studies that investigate the use of drama or drama-related procedures in the treatment of certain psychopathologies or mental illness and anxiety and post-Covid psychological stress and trauma.  The authors did collect studies within certain parameters.  There are not a lot of studies to begin with, as pointed out by the authors towards the end of the article.  What articles they used for the meta-analysis seemed sufficient.  In general, the resulting analysis point to the positive utility of drama and drama-related procedures of program in alleviating or addressing stress, anxiety, depression, and other psychopathologies emerging during post-Covid years.  This is good.  However, it would be a lot more useful if some sort of list of common denominators is presented or explained or theorized that indicate what it is about drama that makes it work for those in trauma or in some state of mental illness during and after the main Covid years.  What aspect or critical aspects of drama make this work?  The authors stated that communication clarity is one.  But how is that different from the non-drama therapies or programs that also focus on communication strategies and is also effective to a degree in addressing psychological anxiety or stress?  The entire article is clear in announcing how great drama is as a tool for these mental health scenarios.  However, a finer dissection of drama should be proposed and made to examine why it works and how it works. The authors should expand on this proposal and offer it as a challenge in the next step of utilizing drama as a therapeutic tool during post-Covid times.

Round 2

Reviewer 2 Report

I would like to thank the authors for considering my suggestions and implementing the required changes to the manuscript. I am satisfied that the paper can now be accepted for publication.

Author Response

Dear Reviewer,

Thank you so much for your kind words. We are delighted to hear that the revision has met with your approval. We greatly appreciate you taking the time to review the paper and providing the constructive feedback, which was extremely valuable to us. Sincerely wish you the best.